# Refining Monte Carlo Tree Search Agents by Monte Carlo Tree Search

## Abstract

Reinforcement learning methods that continuously learn neural networks by episode generation with game tree search have been successful in two-person complete information deterministic games such as chess, shogi, and Go. However, there are only reports of practical cases and there are little evidence to guarantee the stability and the final performance of learning process. In this research, the coordination of episode generation was focused on. By means of regarding the entire system as game tree search, the new method can handle the trade-off between exploitation and exploration during episode generation. The experiments with a small problem showed that it had robust performance compared to the existing method, Alpha Zero.

## 1 Introduction

The result that computer programs beat professional human players on chess, shogi and Go was a huge achievement in computer science. In particular, the development of highly general methods totally changed our perspective about two-person complete information deterministic games.

Then, has this field already "finished"? My answer is no. To deal with many games, more robust methods are required to free humans from hyperparameter tuning. Moreover, the challenge to the god of games won't be finished and we want algorithms that can achieve better final performance.

This study attempts to bring suggestions for recent achievements in two-player complete information deterministic games from classical "game tree search" context. More specifically, this is a new approach in which the reinforcement learning system using game tree search itself is handled as game tree search.

## 2 Approach to complete information games

### 2.1 Reinforcement Learning

Turn-based complete information environments this report deals with is called Markov decision process. When the state $s_t$ is observed at the time $t$ and the action $a_t$ is performed, the environment returns immediate reward $r_t$ and the next state $s_{t+1}$.

In general, reinforcement learning is a framework which improves agent's policy $p$ and value estimation $v$ in a given environment. While $p$ and $v$ can be stored in a table for all states in a small environment, function approximation is commonly used for larger problems. Model-free reinforcement learning using a deep neural network (DNN) has been achieving great resutls after the success on several video game environments.

This study is aimed at two-person complete information deterministic games. While various model-free reinforcement learning algorithms can be applied to these games, we can also apply model-based reinforcement learning because agents can keep the state transition model of the environment.

### 2.2 Game Tree Search

In particular, we can apply forward planning to the two-person complete information deterministic game using the fact that the progress of the game is represented as a tree structure. Ideally, the

winning player and the optimal action in any two-player complete information deterministic game in which a finite number of states can appear can be determined by performing minimax search on the game tree within finite time.

Actually, it is not necessary to search all states in order to determine the winner at the initial state. We can use alpha-beta search which reduces useless search. However, if the game is not very easy, it is not realistic to complete alpha-beta search that takes exponential time against the size of the game.

For this reason, various methods have been developed so far that can be executed in a realistic time and achieve sufficient performance. Roughly, both the algorithmic improvement of the search and the reduction of the search amount by the function approximation have been much effective.

For example, Monte Carlo tree search (MCTS), which is the base of this research, performs effective forward search using function approximations. With the good property that the performance improves monotonously the the time being spent is, MCTS is widely used in this field.

## 2.3 MONTE CARLO TREE SEARCH

There are a number of variants of Monte Carlo tree search, but the algorithm PUCT in algorithm 1 2 is described in this report. In that pseudo code, it is assumed that the turn player is changed every turn, and the discount factor is 1, and the immediate reward is obtained only at the terminal state.

The Monte Carlo tree search starts from an empty game-tree, and the new node which represents one state is added to expand the game tree in each simulation. After expanding tree, the new state value which approximation function (e.g. neural nets) has returned is fed back to the upper nodes to update the estimated action value of each node.

The core of the Monte Carlo tree search is how to decide action response on the game tree, where the bandit algorithm is used.

The bandit algorithm is an algorithm for maximizing the cumulative reward in a problem setting where we select one of multiple slot machines that we do not know the true average reward in one turn. It is required for bandit algorithms to handle the trade-off between exploration and exploitation.

In the Monte Carlo tree search, by using this bandit algorithm for selection phase at each node, the best action is asymptotically proven and selected at each node, and as a result, it can asymptotically find the optimal action. PUCT uses the following PUCB formula 1 as the bandit algorithm.

$$action \leftarrow \underset{a \in LegalActions(s)}{\arg\max} (\mathbf{q}_{s,a} + C\mathbf{p}_{s,a} \frac{\sqrt{\sum_b \mathbf{n}_{s,b}}}{\mathbf{n}_{s,a} + 1}) \tag{1}$$

In previous research, the output of Monte Carlo tree search usually only consists the best action or the new policy probability distribution at the root node. However, the new estimated value at the root node ($v_s$ in algorithm 2) is also returned in algorithm 2 in order to use it in this research.

Although it is a detailed point, there is important point that is heavily related to the experimental results of this report, which is that some noise is added to the policy by applying AddNoise() function only at the root node shown in the algorithm 1. This is because the root node is recognized not only as bandits but also as best arm identification problems. Therefore the action selection should be more exploratory to find good actions which have been evaluated poorly.

In previous method, noises generated by the Dirichlet distribution are added as described in 2. This study follows that.

$$\mathbf{p} \longleftarrow (1 - \epsilon) \, \mathbf{p} + \epsilon \, \mathrm{Dirichlet}(\alpha) \tag{2}$$

## 2.4 REINFORCEMENT LEARNING WITH MONTE CARLO TREE SEARCH

Also from the viewpoint of reinforcement learning, it is useful for agents to be able to perform the game tree search using the true environment model. This is because agents can usually obtain a

---

**Algorithm 1:** Simulation in PUCT

---

**Data:** state $s$, neural net $net$, depth $d$
**Result:** reward or new q-value
**begin**
    **if** $s$ is terminal state **then**
        | **return** reward at $s$
    **end**
    **if** $n_{s,a} = 0$ for all $a \in \text{LegalActions}(s)$ **then**
        | $\mathbf{p}_s, v_s \longleftarrow net(s)$
        | **return** $v_s$
    **end**
    $\mathbf{p} \longleftarrow \mathbf{p}_s$
    **if** $d = 0$ **then**
        | $\mathbf{p} \longleftarrow \text{AddNoise}(\mathbf{p})$
    **end**
    $action \longleftarrow$ select by $\text{PUCB}(\mathbf{n}_s, \mathbf{q}_s, \mathbf{p})$
    $s' \longleftarrow \text{Transition}(s, action)$
    $q_{new} \longleftarrow -\text{Simulation}(s', d + 1)$
    $q_{s,a} \longleftarrow (q_{s,a} + q_{new})/(n_{s,a} + 1)$
    $n_{s,a} \longleftarrow n_{s,a} + 1$
    **return** $q_{new}$
**end**

---

**Algorithm 2:** PUCT

---

**Data:** state $s_0$, neural net $net$, total simulation count $N$
**Result:** posterior estimation of policy and value at $s_0$
**begin**
    **while** $\sum \mathbf{n}_{s_0} < N$ **do**
        | $\text{Simulation}(s_0, net, 0)$
    **end**
    $\mathbf{p}_{post} \longleftarrow \text{Posterior}(\mathbf{n}_{s_0}, \mathbf{q}_{s_0}, \mathbf{p}_{s_0})$
    $v_{post} \longleftarrow \mathbf{p}_{post} \cdot \mathbf{q}_{s_0}$
    **return** $\mathbf{p}_{post}, v_{post}$
**end**

---

better strategy and a more accurate value estimation by performing the game tree search. Therefore, the game tree search can be seen as a kind of operator to improve there estimates.

Various reinforcement learning studies using this property have been done. In particular Alpha Zero and EXpert ITeration are the great success, which enables us to handle various games. These algorithms continuously update approximation function by making training targets by using Monte Carlo tree search as such operator.

The procedure in Alpha Zero is followed in this research. The algorithm 3 describes the single thread version of Alpha Zero.

---

**Algorithm 3:** Alpha Zero (single thread version)

---

**Data:** total epochs $E$, games per epoch $G$, each simulation count $N$
**Result:** trained neural net
**begin**
    $net \longleftarrow$ initialized neural net
    $episodes \longleftarrow \emptyset$
    **for** $e \in 1..E$ **do**
        **for** $g \in 1..G$ **do**
            $traj \longleftarrow \emptyset$
            $s \longleftarrow$ original state $s_0$
            **while** $s$ is not terminal state **do**
                $\mathbf{p}_s, v_s \longleftarrow \text{PUCT}(s, net, N)$
                $action \longleftarrow$ sampled from $\mathbf{p}_s$
                $traj \longleftarrow traj \cup (\mathbf{p}_s, v_s, action)$
                $s' \longleftarrow \text{Transition}(s, action)$
            **end**
            $r \longleftarrow$ reward at $s$
            $episodes \longleftarrow episodes \cup (traj, r)$
        **end**
        $net \longleftarrow \text{TrainNeuralNet}(net, episodes)$
    **end**
    **return** $net$
**end**

---

# 3 REINFORCEMENT LEARNING WITH MASTER GAME TREE

In the previous method, the variety of generated episodes is dependent on random action selection after Monte Carlo tree search.

However, if there is a state where the policy calculated by the neural network is too sharp, it might be difficult to reverse action evaluation in a limited number of simulations. Therefore, there may be is several weak states even after long-time learning process. The experimental results in this study suggest that such phenomena can be actually observed.

Wo (2019) proposed the preprocess and the post-process to the simulation counts in order to improve explorability at the root node. His result suggested that this method can accelarate the learning speed in leaning Go.

In this report, from the other perspective, a method to apply Monte Carlo tree search to the whole episode generation procedure is proposed.

## 3.1 MASTER GAME TREE

The output of the Monte Carlo tree search is a new probability distribution and a new state value estimation. Therefore, the Monte Carlo tree search itself can be also regarded as a function of policy and value estimation.

The fact that the Monte Carlo tree search itself can be regarded as a function of policy and value estimation implies that we can execute Monte Carlo tree search which uses this function. In the proposed method, the Monte Carlo tree search is performed on the master side that organizes episode generation. This is hereinafter referred as a **master game tree**.

The value of the table 1 is stored in each node of the master game tree.

Table 1: Values stored in ordinal game tree and master game tree

| variable | ordinal game tree | master game tree |
|---|---|---|
| $p_{s,a}$ | policy by DNN | policy by MCTS |
| $v_s$ | value by DNN | value by MCTS |
| $n_{s,a}$ | the number of simulations | the number of episodes |
| $q_{s,a}$ | estimated action value in simulations | estimated action value in episodes |

Thus, the values stored in each node in the master game tree has a one-to-one correspondence with the values stored in the ordinal Monte Carlo tree search node. Therefore the master game tree can be implemented as same as the ordinal one.

It is desired that the master game tree converges to the optimal strategy asymptotically by assigning episode generation by this master game tree. It seems that theoretical convergence will be guaranteed if same classic bandit algorithm with no prior information is used in each action selection on the master game tree.

However, through the success of research on the Monte Carlo tree search so far, function approximation is also used on the master game tree side in the proposed implementation in this report. It means that PUCT is also applied to the master game tree.

Since the proposed method uses the master game tree so as to refine the performance of Monte Carlo tree search by the other Monte Carlo tree search, this report named it **MbM** (**MCTS-by-MCTS**).

## 3.2 UPDATE POLICY AND VALUE IN THE MASTER GAME TREE

The difference between this master game tree search and the ordinal Monte Carlo tree search is that the accuracy of the policy and the value estimation returned by neural nets is increased through proceeding training. Therefore, by updating the $p$ and $v$, which were fixed after the node creation timing in ordinal PUCT, bandits on the nodes that has already created can behave as if they had created recently.

Instead of replacing $p$ or $v$ with new ones, the weighted average between old ones and new ones are calculated after each episode in the proposed method. It helps absorb the fluctuation of the result of each trial of the Monte Carlo tree search. This weighting ratio can be increased as the generation of neural networks progresses to emphasize newer estimation. In this report, however, a simple average is used.

Specifically, the master game tree is updated as in the algorithm 4. The state value estimate of the newly added node and the difference between the old and the new state value estimation inside the tree are summed and propagated to the upper nodes.

Using this update rule, the entire proposed method is discribed in the algorithm 5. The NextPathBy-MasterTree() function, which generates the opening action sequence by descending the master game tree, is almost the same as the operation of descending the tree in the algorithm 1. When going down the master game tree, the noise is always added to $\mathbf{p}_s$ in the algorithm 1 in the regular version of the proposed method. This point will be discussed at the later part of this report.

---

**Algorithm 4:** Master Game Tree Update

---

**Data:** state $s$, guide path $path$, trajectory (sequence of (policy, value, action) tuple) $traj$,
    reward $r$

**Result:** reward or new q-value with internal tree updates

**begin**
  **if** $|path| = 0$ **then**
    **if** $s$ *is terminal state* **then**
      **return** $r$
    **end**
    $\mathbf{p}_s, v_s, action \longleftarrow traj[0]$
    **return** $v_s$
  **end**
  $\mathbf{p}_{snew}, v_{snew}, action = traj[0]$
  $w \longleftarrow 1/\sum_{a \in \text{LegalActions}(s)} n_{s,a}$
  $v_{sold} \longleftarrow v_s$
  $\mathbf{p}_s \longleftarrow \mathbf{p}_s * (1 - w) + \mathbf{p}_{snew} * w$
  $v_s \longleftarrow v_s * (1 - w) + v_{snew} * w$
  $s' \longleftarrow \text{Transition}(s, action)$
  $q_{new} = -\text{UpdateMaster}(s', path[1:], traj[1:], r)$
  $q_{s,a} \longleftarrow (q_{s,a} + q_{new})/(n_{s,a} + 1)$
  $n_{s,a} \longleftarrow n_{s,a} + 1$
  **return** $q_{new} + v_s - v_{sold}$
**end**

---

**Algorithm 5:** Proposed Method: MbM (single thread version)

---

**Data:** total epochs $E$, games per epoch $G$, each simulation count $N$

**Result:** trained neural net

**begin**
  $net \longleftarrow$ initialized neural net
  $episodes \longleftarrow \emptyset$
  **for** $e \in 1..E$ **do**
    **for** $g \in 1..G$ **do**
      $path \longleftarrow \text{NextPathByMasterTree}()$
      $traj \longleftarrow \emptyset$
      $s \longleftarrow$ original state $s_0$
      **while** $s$ is not terminal state **do**
        $\mathbf{p}_s, v_s \longleftarrow \text{PUCT}(s, net, N)$
        **if** $|traj| < |path|$ **then**
          $action \longleftarrow path[|traj|]$
        **else**
          $action \longleftarrow$ sampled from $\mathbf{p}_s$
        **end**
        $traj \longleftarrow traj \cup (\mathbf{p}_s, v_s, action)$
        $s' \longleftarrow \text{Transition}(s, action)$
      **end**
      $r \longleftarrow$ reward at $s$
      $episodes \longleftarrow episodes \cup (traj, r)$
      $\text{UpdateMaster}(s_0, path, traj, r)$
    **end**
    $net \longleftarrow \text{TrainNeuralNet}(net, episodes)$
  **end**
  **return** $net$
**end**

---

## 4 EXPERIMENTS

### 4.1 COMPARISON IN VARIOUS GENERATION TEMPERATURE

In the experiment, in the tic-tac-toe where the solution is already sought for the small size, the existing method AlphaZero and the proposed method learned for 20,000 battles, The following points were compared by the obtained neural network policy.

- Ratio of defeating against random players
- Ratio of defeating to perfect players

Tic-tac-toe is obviously very simple task, and it is easy to obtain a perfect strategy by minimax search. Only by function approximation using neural network, however, it may be necessary to generate a certain number of episodes and train from them to obtain good policy. Still, it is so easy problem that we desire that neural nets can output the optimal strategy.

The experiments were performed with the following settings.

In the proposed method, in the bandit on the master game tree side, the same as the root node on the episode generation side. We tried two ways of adding Dirichlet noise to the policy and not adding it.

Especially when adding Dirichlet noise to the policy, if the other parameters are the same in Alpha Zero and the MbM, episode generation become more exploratory in the proposed method than Alpha Zero. The performance can vary greatly depending on the effect (actually shown in the experimental results in this paper).

Therefore, the initial values of episode generation temperature parameters were varied as 0.3, 0.6, 1.2, and 2.4, respectively into both the previous method and the proposed method.

Other experimental settings and hyperparameters are shown below. For details, please refer to the online experimental code. [1].

- attenuation rate of generation temperature: 0.8
- $C$ in PUCB: 2.0
- noise to the policy $\alpha = 0.1, \epsilon = 0.25$
- the number of epochs: 200
- the number of episodes per epoch: 100
- input feature: 2 planes (the position mine, the position of opponents)
- detail of neural nets: (3x3 convolution with 32 filters + BN), (Wide Resnet x 4 layers), (1x1 convolution with 2 filters + BN + fully connected + softmax in policy head), (1x1 convolution with 2filters + BN + fully connected + tanh)
- detail of training data: (batch size 32 x the number of episodes before x 50 loop) per epoch,
- detail of optimization: SGD (learning rate=1e-3, momentum=0.75 weight decay=1e-4)

The experimental results are plotted in the figure 1 2, comparing the temperature conditions for each opponent (random player, perfect player) in the previous method AlphaZero and the proposed method MbM.

From the result of AlphaZero in the FIg. 1 2, Alpha Zero has not reached the optimal strategy within 200 epochs when the temperature of episode generation is low. In comparison, the result of MbM in Fig. 1 2 shows that almost optimal strategies seem to be obtained under all conditions.

Compared through those results, the performance of proposed methods are equal to or better than the best one of AlphaZero.

---

[1]https://github.com/advancedailab/RecursivePlanning

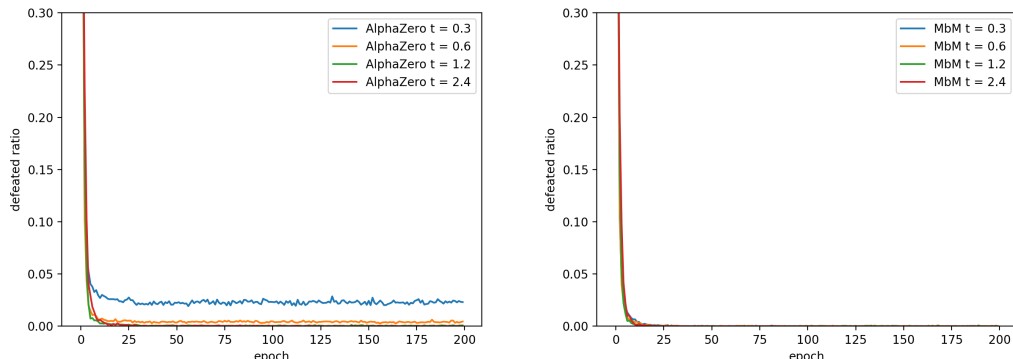

Figure 1: Match results against random player

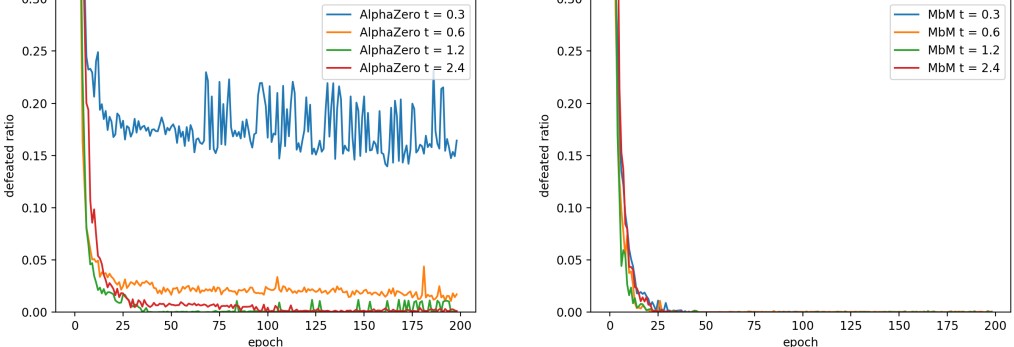

Figure 2: Match results against perfect player

## 4.2 Effect of noise added policies in master game tree

In the result of the previous experiment, the proposed method MbM showed good performance. However, compared to the result of Alpha Zero, there seems to be the possibility that the difference of performance caused only by the explorability of episode generation by adding noise to the policy when going down the master game tree.

Additional experiment was done with the following two modified versions of the proposed method under the temperature is low ($t = 0.3$ or $t = 0.6$).

- MbM-NoNoise: no noise is added to the policy
- MbM-Relaxation: applying conversion to the policy in order to soften sharpness of it.

In the latter one, the policy $\mathbf{p}$ is modified as $\mathbf{p} \longleftarrow (\mathbf{p} + 0.1)/(\sum(\mathbf{p} + 0.1))$. The figure shows learning up to 100 epochs under each condition and comparing the results with those in previous experiment.

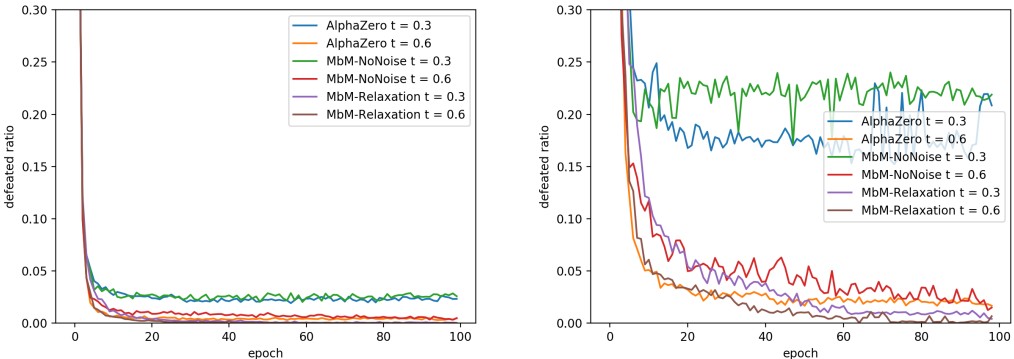

Figure 3: Comparison between modified version of MbM

The results in Fig. 3 show that MbM-NoNoise, which does not add noise, is even worse than Alpha Zero. MbM-UniformRelaxation, which the fixed amount of diversity is added to the policy, was not so good as MbM but pretty better than Alpha Zero. This point will be discussed in the final section.

## 5 Discussion

In this study, we examined a very simple task, Tic-tac-toe. First of all, it was shown that obtaining the optimal strategy is sometimes difficult depending on the parameters.

The results suggest that reinforcement learning methods like Alpha Zero often suffer from naiveness against exploration.

In the proposed method, it is possible to vary the beginning of the game in the episode generation by the master game tree. The results suggest that the proposed method has ability to control the search for the beginning of the game by adding proper noise.

On the other hand, when PUCT using the strategy as it is applied to the master game tree (MbM-NoNoise), the performance was lower than the baseline. The reason of this result is that the policy has converged earlier than the effect of exploration in the master game tree. Due to this point, it was not effective.

In this report, PUCT is applied to the master game tree as same as ordinal game tree. However, it is necessary to examine a mechanism that makes it more exploratory.

Lastly, in this study, we verified only one of the simplest games, Tic-tac-toe. From the experimental results in this paper, it is expected that the proposed method can produce robust results with respect

to temperature parameters even for larger games. It will also be necessary to verify whether the speed of improvement in real time is better that previous methods.

I hope that the combination of tree search and reinforcement learning will be used for a wider range of domains if there exists the method in which both stableness and speed are better performance.

