# OpenReview forum: "REFINING MONTE CARLO TREE SEARCH AGENTS BY MONTE CARLO TREE SEARCH"
_ICLR.cc/2020/Conference — Reject_

### Official Review · AnonReviewer3 · 2019-10-21
**Official Blind Review #3**

**Rating:** 1

**Review:**

This paper appears to be attempting to improve Monte Carlo tree search, though it's not clear how. The paper is very poorly written, with many incomprehensible passages. Moreover there are no references at all. It does not appear to be a serious academic endeavor, merely a set of notes. There are many mathematical symbols not defined, much of the paper consists of blank space yet it goes up to 10 pages, the algorithm boxes are very difficult to follow, the writing is so poor that I cannot decipher what the intent even was. A significant amount of work would be required to make this acceptable for publication in any venue.

**Experience Assessment:**

I have published in this field for several years.

**Review Assessment: Checking Correctness Of Derivations And Theory:**

N/A

**Review Assessment: Checking Correctness Of Experiments:**

N/A

**Review Assessment: Thoroughness In Paper Reading:**

N/A

---

> ### Author Response · Authors · 2019-11-06
> **Thank you for your review**
>
> Thank you deeply for your reviewing my paper.
> I am so sorry that even the proposed method itself is hard to be understood.
> I hope I can write better one in the next time.

---

### Official Review · AnonReviewer2 · 2019-10-22
**Official Blind Review #2**

**Rating:** 1

**Review:**

Badly and unprofessionally written

I had and still have a hard time to report what this paper is about. Generally, I can tell probably this paper study the Monte Carlo Tree Search-based methods in the field of Reinforcement Learning.

The paper is badly written with grammatical issues almost in all the sentences. I definitely urge the author(s) to take the writing significantly more seriously. I almost all the time do not end up rejecting a paper due to the lack of clear writing, but for this paper, unfortunately, despite spending time, I could not understand the paper.


Comments beside the grammatical issues:
I am sorry but I could not understand the abstract.
What does the author(s) mean by "In this research, the
coordination of episode generation was focused on." Same issue with almost all the sentences in the abstract. Unfortunately, I could not follow any of them. Moreover, this paper is full of inaccurate and unjustified statements. Furthermore, again, unfortunately, this paper does not represent even the preliminary elements of scientific writing.


I decided to provide further comments, but sadly I had to change my mind because otherwise, I had to write at least one paragraph of explanation for each paragraph of this paper. With that regard, I do not think it is an appropriate way to utilize the review process. But I would be happy to provide more abstract comments upon request.

**Experience Assessment:**

I have published in this field for several years.

**Review Assessment: Checking Correctness Of Derivations And Theory:**

I did not assess the derivations or theory.

**Review Assessment: Checking Correctness Of Experiments:**

I did not assess the experiments.

**Review Assessment: Thoroughness In Paper Reading:**

I read the paper thoroughly.

---

> ### Author Response · Authors · 2019-11-06
> **Thank you for your review**
>
> Thank you deeply for your reviewing my paper and for your kindness.
> I want to improve my paper writing skills and I hope to earn more money to feel free to ask for proofreading before the next chance.
>
> I hope to know whether there has been other publishing for this topic (meta procedure for MCTS based reinforcement learning).
> After registration of this paper, I finally found the almost same idea (meta-tree) in my country, but I could not find international papers so far.
> Though AlphaZero itself was amazing algorithm, I think there might be several possible updates for robustness or speedup.

---

### Official Review · AnonReviewer1 · 2019-10-22
**Official Blind Review #1**

**Rating:** 1

**Review:**

This paper tries to improve exploration performed by alphazero in a game of tic-tac-toe using MCTS. The paper proposes to use master game tree (which is MCTS as well) to control the generation of episodes when solving a game using MCTS.
This is a clear case of less than half-baked paper. The paper does not cite any previous research (no references) and is poorly written. I believe this is sufficient ground for not recommending accept. I would suggest the authors to re-submit when the paper is complete, maybe, start with related work and references.

**Experience Assessment:**

I have read many papers in this area.

**Review Assessment: Checking Correctness Of Derivations And Theory:**

I did not assess the derivations or theory.

**Review Assessment: Checking Correctness Of Experiments:**

I did not assess the experiments.

**Review Assessment: Thoroughness In Paper Reading:**

I made a quick assessment of this paper.

---

> ### Author Response · Authors · 2019-11-06
> **Thank you for your review**
>
> Thank you deeply for your reviewing my paper.
> I didn't know how to use .bib file, then I had no choice but to remove all references.
> I would like to improve my paper writing skills and to do more research on this topic.

---

### Decision · Program_Chairs · 2019-12-19

**Decision:**

Reject

**Comment:**

This paper is a clear reject. The paper is very poorly written and contains zero citations. Also, the reviewers have a hard time understanding what the paper is about.